# Psychosocial career preoccupation and organisational commitment at higher educational institutions in Ghana: The role of workplace friendship

Dorothy Amfo-Antiri[1], Isaac Tetteh Kwao[2]*, Kassimu Issau[3], Emmanuel Essandoh[4], Emmanuel Agyenim Boateng[5], Esther Bema Nimo[6]

1 Department of Human Resource Management, School of Business, University of Cape Coast, Cape Coast - Ghana, 2 Department of Human Resource Management, School of Business, University of Cape Coast, 3 Department of Marketing and Supply Chain management, School of Business, University of Cape Coast, 4 Department of Human Resource Management, School of Business, University of Cape Coast, 5 Department of Human Resource Management, School of Business, University of Cape Coast, 6 School of Pharmacy and Pharceutical Sciences, University of Cape Coast

* isaac.kwao@ucc.edu.gh

## Abstract

Empirically, the study examined how psychosocial career preoccupation [PCP] directly affects organizational commitment [OC] and indirectly affects the same via workplace friendship [WF] among administrative and teaching staff in public higher education institutions in Ghana. The numerical orientation to the data analysis supported the use of the explanatory research design and quantitative approach. The population included both administrative and teaching staff in public higher education institutions in Ghana with at least one year of service. Structured questionnaires were issued for the primary data collection via the convenience sampling technique and the response rate was 96% (288 valid responses from 300 approached participants). The unit of analysis was at the individual level. SMART-PLS software (version 4.1.0.8) was used for the data processing. The repeated indicator structural modelling technique was employed to configure the model reflectively to test the hypotheses. PCP fails to contribute significantly to predicting OC directly ($\beta = 0.045$, $p = 0.250$). However, PCP contributes significantly in a strong manner to predicting WF ($\beta = 0.454$, $p < 0.001$), and WF significantly predicts OC ($\beta = 0.442$, $p < 0.001$). WF successfully mediates the predictive relationship between PCP and OC with a significant indirect effect ($\beta = 0.201$, $p < 0.001$). Higher education policymakers in Ghana are advised to develop and enforce policies that promote the integration of career development with social support systems within institutions, mandating structured programs that address employees' career advancement while actively encouraging workplace friendship networks. It is advisable for higher education administrators and human resource practitioners in Ghana to prioritize cultivating workplace friendships as a strategic resource to enhance organisational commitment while implementing

**Data availability statement:** All data for analysis has been provided in the paper (see appendix 1).

**Funding:** The author(s) received no specific funding for this work.

**Competing interests:** We do not have competing interest.

comprehensive career development programs that address psychosocial career preoccupations. These results provide an innovative, contextually relevant perspective on the mediating role of workplace friendship in translating career concerns into organisational commitment within Ghanaian higher education, thus enhancing current theoretical frameworks by demonstrating the indirect pathway through social relationships rather than direct effects of career preoccupations.

## Introduction

Organisational commitment has emerged as a critical determinant of institutional sustainability in today's volatile employment landscape [1,2]. Organisational commitment represents employees' psychological attachment to their organisation, which significantly influences retention, performance, and citizenship behaviors [3,4]. Concurrently, psychosocial career preoccupations, comprising career establishment preoccupations, career adaptation preoccupations, and work-life adjustment preoccupations, reflect employees' evolving career-related concerns throughout their professional journey [5,6]. Contemporary institutions face mounting pressure to address these psychosocial career dimensions as they substantially impact employee engagement, productivity, and institutional effectiveness [7]. Research indicates that institutions responsive to employees' career preoccupations report higher retention rates and improved performance outcomes [8–10].

Research on psychosocial career preoccupations and organisational commitment has predominantly examined corporate environments with limited investigation in Higher Educational Institutions (HEIs), particularly within developing economies like Ghana. Ghanaian HEIs confront unprecedented challenges including resource constraints, surging enrollment demands, internationalization pressures, and intense competition for academic rankings [11,12]. The interplay between psychosocial career preoccupations and organizational commitment carries significant implications for Ghanaian HEIs, where quality education delivery and research output depend heavily on faculty and staff dedication [6,13]. Human resource management (HRM) in Ghanaian higher education is increasingly complex due to funding uncertainties, talent retention challenges, and rising stakeholder expectations. These challenges are compounded by the need for institutions to adapt to evolving educational landscapes, regulatory environments, and technological advancements [14].

Empirical investigations have yielded inconsistent findings regarding the relationship between psychosocial career preoccupations and organisational commitment across organisational contexts. Recent studies revealed positive correlations between certain dimensions of psychosocial career preoccupations and organisational commitment. For instance, Career adaptation preoccupations, which involve adjusting to career changes and challenges, have been shown to significantly predict normative commitment. This indicates that individuals who are adaptable in their careers may feel a moral obligation to remain with their organization [15,16].

However, career establishment preoccupations are associated with a negative relationship with affective commitment, indicating that individuals focused on establishing their careers may feel less emotionally attached to their organisations. However, these preoccupations positively relate to continuance commitment, suggesting that such individuals may remain with an organisation due to perceived costs of leaving or lack of alternatives [15]. Work-life adjustment preoccupations on the other hand have varying impacts on different dimensions of organisational commitment. For example, in educational settings, these preoccupations can influence affective, continuance, and normative commitments differently, depending on the individual's ability to balance work and personal life [17]. Workplace friendship has gained prominence as a potential influencing factor in organisational behavior research. Studies indicate that workplace friendship can enhance job embeddedness, organisational commitment, and attenuate turnover intentions [18,19]. Despite growing interest in career dynamics and organisational commitment, several critical gaps persist in the literature. The existing studies [6,7] primarily focus on the relationship between psychosocial career preoccupations and organisational commitment, but they often do not delve deeply into the multidimensional aspects of these preoccupations. Studies indicate that psychosocial career preoccupations can significantly influence organisational commitment, but the complexity and multidimensionality of these preoccupations are not fully explored in the context of organisational commitment. Workplace friendship has been recognized as a significant factor influencing organisational outcomes, including organisational commitment. However, its role as a mediator between career preoccupations and commitment is less explored. The existing literature [19, 20] suggest that workplace friendships can enhance organisational commitment by fostering a supportive work environment, which in turn can influence career-related outcomes. Higher education research in Ghana has identified several workplace challenges including difficult colleague interactions, gendered inequities in balancing professional and personal responsibilities [22], and training opportunities perceived as inequitable [23]. Yet there remains a significant research gap regarding how psychosocial career pre-occupation interacts with workplace friendships in higher education institutions.

This study employs social exchange theory [24] and career construction theory [25] as its theoretical foundation. Social exchange theory posits that employment relationships involve reciprocal exchanges where employees develop commitment when organisations fulfill their professional and psychological needs [26]. Career construction theory contends that individuals actively construct their careers through adaptive behaviors responding to work environment demands and personal preoccupations [27]. These complementary perspectives have been extensively applied in organisational behavior and career development research.

This research makes a significant theoretical contribution by explaining the mechanisms through which psychosocial career preoccupations influence organisational commitment in higher education, with workplace friendship serving as a mediator – illuminating the underlying social processes that connect these constructs. Additionally, the study offers practical contributions by providing empirical evidence from an underrepresented African educational context, enhancing generalizability of existing frameworks. The findings deliver actionable insights for higher education administrators and human resource practitioners in Ghana regarding strategies to strengthen organisational commitment through addressing career preoccupations and fostering workplace relationships. Furthermore, this research aligns with Sustainable Development Goals (SDGs) 4 (Quality Education), 8 (Decent Work and Economic Growth), and 10 (Reduced Inequalities) by promoting supportive work environments that enhance educational quality, foster sustainable employment practices, and address workplace inequities in Ghanaian higher education institutions.

The remainder of this paper is organised as follows. First, we provide a contextual background of Ghanaian higher educational institutions. This is followed by a comprehensive review of literature and development of the conceptual framework and research hypotheses. Next, we detail the methodology employed in the study, including sampling procedures, measurement instruments, and analytical approaches. We then present the results of data analyses and hypotheses testing, followed by a thorough discussion of theoretical and practical implications. Finally, we acknowledge limitations of the current investigation and suggest promising avenues for future research.

## Literature review and hypothesis development

### Psychosocial Career Preoccupation (PCP)

[5] conceptualized psychosocial career preoccupations as career concerns that individuals experience at particular life points, influencing their career decisions and behaviors. Psychosocial career preoccupations comprise three dimensions: career establishment preoccupations (concerns about fitting into groups, self-expression opportunities, economic stability, and advancement), career adaptation preoccupations (concerns about career changes, adapting to changing contexts, and employability security), and work-life adjustment preoccupations (concerns about balancing work with personal responsibilities) [6,15]. Unlike traditional career stages, these preoccupations focus on internal cognitive-affective concerns that may occur across different age groups rather than being time-bound [7].

In higher educational institutions facing turbulent and competitive environments, psychosocial career preoccupations significantly influence employee behaviors and attitudes [28]. Individuals with high career preoccupations seek new opportunities, take calculated risks, and actively shape their careers instead of merely adapting to circumstances [16]. As noted by [9], these preoccupations extend beyond traditional career orientations, reflecting the need to address the fast-paced and ever-changing higher education environment that requires transformative, innovative, and proactive approaches to career management.

### Organisational Commitment (OC)

According to [1], organisational commitment is operationally characterized as a psychological state that binds an individual to the organisation. Meyer and Allen's three-component model of organisational commitment has been widely adopted in research and includes affective commitment, continuance commitment, and normative commitment [2,29]. Affective commitment refers to employees' emotional attachment to, identification with, and involvement in the organisation. This dimension represents a desire-based attachment where employees stay with the organisation because they want to do so [13]. Employees with strong affective commitment identify with the organisation's goals and values, participate in organisational activities, and desire to maintain organisational membership. Continuance commitment refers to employees' awareness of the costs associated with leaving the organisation. This dimension represents a need-based attachment where employees stay because they need to do so, often due to financial considerations or limited alternatives [12]. Employees with high continuance commitment remain with the organisation because they perceive the costs of leaving as higher than the benefits.

Normative commitment reflects employees' feelings of obligation to continue employment. This dimension represents an obligation-based attachment where employees stay because they feel they ought to do so [30]. Employees with strong normative commitment feel a sense of duty or moral obligation to remain with their organisation, often influenced by organisational socialization or benefits received. In higher educational institutions, organisational commitment has been associated with improved teaching quality, research productivity, and institutional effectiveness [14]. Faculty and staff with high levels of organisational commitment demonstrate greater willingness to engage in discretionary behaviors that benefit the institution beyond formal job requirements [11].

### Workplace Friendship (WF)

Workplace friendship is defined as a voluntary workplace relationship that includes mutual trust, commitment, reciprocal liking, and shared interests or values [20]. It has been operationalized into two dimensions: friendship opportunity and friendship prevalence. Friendship opportunity refers to the perceived possibility of developing friendships in the workplace. This dimension encompasses organisational and structural factors that facilitate or hinder the formation of workplace friendships, including physical proximity, task interdependence, shared goals, communication channels, and organisational climate [21]. Organisations that provide ample opportunities for social interaction, teamwork, and collaboration tend to foster environments conducive to friendship development.

Friendship prevalence refers to the degree to which genuine friendships exist within the workplace. This dimension reflects the actual presence and quality of friendships among organisational members rather than merely the opportunity to form such relationships [16]. Friendship prevalence is characterized by mutual trust, self-disclosure, voluntary interaction outside work settings, and reciprocal emotional support among colleagues. Research indicates that workplace friendship can enhance job satisfaction, organisational commitment, and overall well-being while reducing stress and turnover intentions [20]. In academic settings, workplace friendships have been found to foster collaborative research initiatives, interdisciplinary teaching approaches, and improved departmental climate [22].

## Psychosocial career preoccupation and organisational commitment

In higher educational institutions, effective career management improves commitment by instilling a positive attitude and culture that aids in achieving career success [7]. According to [6], psychosocial career preoccupations are viable tools utilized by individuals to impact career outcomes. Under competitive situations, different higher educational institutions utilize career development programs to help employees react quickly to environmental changes and achieve organisational and individual objectives. Existing literature on career development identifies different career stages and their positive effect on organisational commitment and management of the organisation [6,30]. Individuals can benefit from career management competencies to deal with the highly turbulent and competitive environment of modern organisations [9]. Career success through proper management can be achieved when employees are encouraged and nurtured [28,31].

The relationship between psychosocial career preoccupations and organisational commitment is grounded in social exchange theory [24], which suggests that employment relationships involve reciprocity where employees develop commitment when organisations fulfill their career needs [26, 31, 32]. Similarly, career construction theory [25] posits that individuals actively construct their careers through adaptive behaviors responding to work environment demands and personal preoccupations [27]. Empirical evidence has demonstrated mixed findings regarding the relationship between psychosocial career preoccupations and organisational commitment. Career establishment preoccupations may be negatively associated with affective commitment as individuals focused on establishing their careers may feel less emotionally attached to their organisations [15]. However, these preoccupations may positively relate to continuance commitment, suggesting that individuals remain with organisations due to perceived costs of leaving or lack of alternatives [30]. Career adaptation preoccupations have been shown to predict normative commitment significantly, indicating that individuals who are adaptable in their careers may feel a moral obligation to remain with their organisation [16]. Work-life adjustment preoccupations have varying impacts on different dimensions of organisational commitment, depending on individuals' ability to balance work and personal life [17].

Based on these theoretical foundations and empirical evidence, the following hypothesis is proposed:

*H1. There is a significant impact of psychosocial career preoccupation on organisational commitment.*

## Mediating role of workplace friendship

Effective career management can act as a critical instrument in fostering workplace relationships and social connections within the organisation, so timely and realistic career decisions can be taken [7, 33], as cited in [6] referred to career management as an enabler that enables individuals to align career behaviors with opportunities, organisational strategies, communicate the best strategies, facilitate the evolution of learning, and promote values of career development. Hence, career management incorporates strategic issues and how an individual defines their career and uses their social resources to strengthen their competencies [23]. Social exchange theory recognizes workplace relationships as the most strategically important of the individual's resources while psychosocial career preoccupation is committed to the discovery and exploitation of strategic value creation [7]. Psychosocial career preoccupation is about taking risks [15] while from a workplace friendship perspective, an organisation's system that fosters experimentation and risk-taking aids learning, relationship creation, and support [19, 20] proposed that successful career management requires proactive workplace

friendship. Their study demonstrated that career management must create an organisational culture in which there is employee cooperation and friendship.

According to social exchange theory, knowledge and support are created, stored, and utilized by individuals and not by organisations as a whole; coordinating and integrating the relationships held by individuals is a difficult task [26]. This can be made possible through psychosocial career preoccupation which is characterized by individuals' ability as integrators and informed decision-makers [6]. Social exchange theory identifies relationships as key resources and strategic assets that empower individuals to create value [20]. Successful utilization of relationship-based resources and effective implementation of workplace friendship is imperative to achieve better career outcomes [19]. Social exchange theory supports the idea that when relationships are effectively managed, they create distinguishing capabilities that contribute to improved performance [22].

From a human capital perspective, individuals with career preoccupations focus on improving knowledge, skills, and capabilities through workplace relationships [7] they respond and innovate by broadening existing networks and skills [6]. According to [20], the career-focused individual looks to workplace friendships for guidance, support, and creating the necessary environment to facilitate innovation, encourage creativity, and facilitate knowledge creation and knowledge sharing. Research has demonstrated that workplace friendship plays a significant role in enhancing commitment, which is used to improve an organisation's performance and to sustain a competitive advantage in the market [21]. Therefore, in today's changing environment, management of workplace relationships is necessary for organisational commitment [19]. Through workplace friendship, individuals look to gain or make potentially useful connections and to make support available to influence organisational commitment [20].

Studies have identified workplace friendship to improve organisational outcomes [19,21]. Managers in higher educational institutions, regardless of their work environment, agree on how important proper relationship management is to organisational commitment [11]. Researchers have indicated ways in which workplace friendship can be protected and utilized to enable individuals to make good decisions using actionable information, experience, and insight [20]. Career construction theory suggests that commitment to effective workplace relationship management in the context of a career strategy is emerging as a potent means of establishing and sustaining competitive advantage [27], ultimately leading to continued organisational commitment. The literature leads to the belief that psychosocial career preoccupation connects with improved workplace friendship that enhances organisational commitment. Accordingly, the study argues that workplace friendship plays a mediating role in the relationship between psychosocial career preoccupation and organisational commitment as it can be envisaged in Fig 1. Based on the literature, it is proposed that:

*H2. There is a significant impact of psychosocial career preoccupation on workplace friendship.*

*H3. There is a significant impact of workplace friendship on organisational commitment.*

## Research methodology

### Design/ Approach/ Sampling/ Data Collection

This study employed an explanatory research design aligned with a post-positivist paradigm, which emphasizes objectivity, empirical testing, and falsifiability [33]. Data were collected using a structured questionnaire designed in English and based on a 5-point Likert scale. While useful for quantifying abstract constructs, such scales may introduce subjective bias [34]. The population comprised full-time administrative and teaching staff at public higher education institutions in Ghana, with participants required to have at least one year of service. Using convenience sampling, 300 participants were approached via departmental emails, personal visits, and official letters distributed through departmental heads. The final valid responses totaled 288, yielding a 96% response rate. To ensure adequate sample size for PLS-SEM, a G*Power analysis recommended a minimum of 305 participants, using parameters: 0.15 effect size, 0.05 error margin and 0.95 power [40]. To address potential non-responses, oversampling was used, and reminders were issued through email and

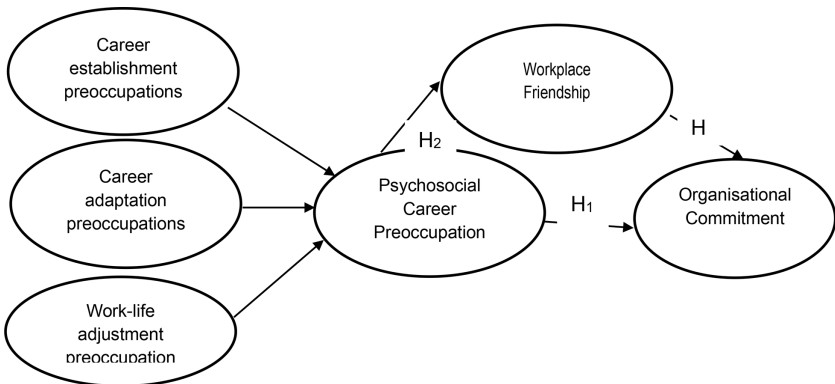

**Fig 1. Conceptual Framework.**

departmental announcements. Data collection occurred over three months (December 2024–February 2025). Demographic details are presented in Table 1 below.

## Ethics statement

Ethical clearance was obtained from institutional authorities, and informed consent was secured from all during the application process and that participation was voluntary. Again, during the administration of the instrument online, a full paged informed consent (written) was attached for the respondent to know that they were at liberty to withdraw from responding at any given point. Considering that the survey content does not involve any sensitive issues and that the data collected are completely anonymous according to the standards.

## Measures and questionnaire development

A structured questionnaire was developed as the primary data collection instrument, utilizing validated scales from previous research. The questionnaire underwent rigorous expert review by five academic specialists (two organisational psychology professors and three human resource management experts) and four administrative heads from Ghanaian public universities to ensure superior content and face validity. The experts confirmed that all construct aspects were adequately covered and that question wording was clear and appropriate for the Ghanaian higher education context. Minor modifications were made based on expert feedback, and the questionnaire was subsequently piloted among administrative and teaching staff at a public university to further refine the survey questions before final administration.

The questionnaire comprised four sections measuring demographic variables, independent variable (psychosocial career preoccupation), mediating variable (workplace friendship), and dependent variable (organisational commitment), all rated on a five-point Likert scale. Psychosocial career preoccupation was measured using 23 items adapted from [5] across three dimensions (career establishment, career adaptation, and work-life adjustment preoccupations), workplace friendship using 12 items adapted from [36] across two dimensions (friendship opportunity and prevalence), and organisational commitment using 23 items adapted from [37] across three dimensions (affective, continuance, and normative commitment). The scales' psychometric properties were rigorously tested for validity and reliability, with items having outer loadings below 0.4 being eliminated following [38] recommendations to improve model performance.

## Analytical Technique

Following questionnaire completion, data retrieval and cleansing were conducted to ensure dataset accuracy by scrutinizing for incomplete responses, missing values, and anomalies. Numerical coding was performed using SPSS software

**Table 1. Demographic Characteristics.**

| Variable | Category | Frequency | Percent (%) |
|---|---|---|---|
| **Age** | 20–30 years | 32 | 11.1 |
| | 31–40 years | 163 | 56.6 |
| | 41–50 years | 81 | 28.1 |
| | 51–60 years | 12 | 4.2 |
| **Gender** | Male | 146 | 50.7 |
| | Female | 142 | 49.3 |
| **Marital Status** | Married | 208 | 72.2 |
| | Single | 70 | 24.3 |
| | Divorce | 3 | 1.0 |
| | Other | 7 | 2.4 |
| **Job Level** | Administrative Assistant | 41 | 14.2 |
| | Senior Administrative Assistant | 70 | 24.3 |
| | Principal Administrative Assistant | 99 | 34.4 |
| | Chief Administrative Assistant | 12 | 4.2 |
| | Technician | 5 | 1.7 |
| | Senior Teaching Associate | 3 | 1.0 |
| | Principal Teaching Associate | 16 | 5.6 |
| | Chief Teaching Associate | 6 | 2.1 |
| | Other | 36 | 12.5 |

(Version 27.0), followed by meticulous data entry to transfer manual data into digital.sav format. The study employed the repeated indicator technique within SMART PLS (version 4.1.0.8) using the PLS-SEM framework to examine hypotheses through a reflectively specified model. A two-stage modeling approach assessed measurement model criteria before evaluating hypothesis significance, with an iterative process eliminating manifest items that did not enhance model quality metrics.

Model specification utilized 5000 subsamples, parallel processing, one-tailed test type, and 0.05 significance threshold with standardized results. PLS-SEM offers distinctive advantages as a comprehensive estimator, providing flexibility and minimizing biases in latent variable model estimation while serving as a pivotal analytical instrument in management and business research [35,39]. It facilitates assessment of causal-predictive complex model robustness and establishes predictive power critical for substantiating explanatory framework relevance and deriving actionable recommendations.

## Model evaluation criteria

The measurement model evaluation encompasses several reliability and validity dimensions with specific threshold criteria. Reliability assessment includes Cronbach's alpha, Rho_a, and composite reliability, all requiring values ≥0.7 according to [40]. Validity measures include convergent validity through average variance extracted (>0.5) and discriminant validity via HTMT ratio (<1) as specified by [41]. Additional quality checks involve common method bias assessment through Inner VIF (<5) and multi-collinearity evaluation using Outer VIF (<5). The structural model evaluation focuses on indicator reliability requiring loadings >0.7 with $p \leq 0.05$, coefficients and effect sizes where $F^2$ values above 0.35, 0.15, and 0.02 represent strong, moderate, and weak effects respectively [42]. The coefficient of determination ($R^2$) interprets results above 0.67 as substantial, 0.33 as moderate, and 0.19 as weak. Robustness testing includes predictive relevance through PLSq²predict (>0) and predictive power via RMSE and MAE metrics [43], alongside Importance-Performance Map Analysis for priority-based quadrant assessment [44,45].

## Common method bias

Structured questionnaire usage presents inherent common method bias (CMB) risk, occurring when similar response requirements analyze dependent and independent latent variables from the same survey [46]. To mitigate CMB risk, the study implemented stringent ex-ante guidelines including reasonable variable numbers for construct measurement, modified pre-validated scales, appropriate sample selection from target population, and negative wording with reverse-coding of select scale items during data processing. Following [47], construct order was modified with stopping clauses added for better respondent comprehension, while scale separation created impressions of distinct constructs, preventing communication inhibition. Reminder calls reduced questionnaire completion errors, and participants unlikely anticipated study questions to purposefully steer responses consistently, as questions formed part of a broader survey. After primary data collection through structured questionnaire, statistical CMB risk magnitude was determined using random dependent methodology, with findings presented in subsequent analysis and as can be seen in Table 2 below.

## Results

### Measurement Model of the initial model

The reliability and convergent validity results from SmartPLS indicate that all constructs demonstrate good internal consistency, with Cronbach's Alpha (CA) values ranging from 0.829 to 0.934, and Composite Reliability (CR) values (both ρ_a and ρ_c) above the acceptable threshold of 0.7, confirming reliable measurement. The Average Variance Extracted (AVE) values mostly exceed the recommended 0.5 threshold, indicating adequate convergent validity for most constructs, though constructs such as OC (0.363), PSCP (0.489), and WF (0.477) show somewhat lower AVE values, suggesting that these constructs may have weaker convergent validity and could benefit from further review or refinement [46,47]. Overall, the measurement model shows acceptable reliability and validity for the study's constructs in Table 3 below

The HTMT (Heterotrait-Monotrait) ratio results indicate the discriminant validity between the constructs. Most construct pairs have HTMT values well below the conservative threshold of 0.85, demonstrating good discriminant validity and confirming that these constructs measure distinct concepts. However, several pairs exceed 0.90 or even surpass 1.0—for example, OC with AC (0.901), OC with CC (0.973), OC with NC (0.960), PSCP with CEP (1.028), PSCP with WLAP (1.029), FO with WF (1.037), and FP with WF (1.057)—which signals potential issues of discriminant validity and suggests these constructs may overlap or measure similar underlying dimensions. These high values imply the need for further investigation, possibly revisiting the measurement items or considering combining constructs to improve model validity [48].

**Table 2. Construct Reliability and Validity.**

| Construct | CA | CR (ρ_a) | CR (ρ_c) | AVE |
|---|---|---|---|---|
| AC | 0.834 | 0.710 | 0.627 | 0.627 |
| CAP | 0.901 | 0.835 | 0.752 | 0.752 |
| CEP | 0.933 | 0.920 | 0.607 | 0.607 |
| CC | 0.863 | 0.804 | 0.558 | 0.558 |
| FO | 0.845 | 0.770 | 0.580 | 0.580 |
| FP | 0.900 | 0.864 | 0.643 | 0.643 |
| NC | 0.905 | 0.862 | 0.704 | 0.704 |
| OC | 0.874 | 0.865 | 0.363 | 0.363 |
| PSCP | 0.934 | 0.927 | 0.489 | 0.489 |
| WLAP | 0.829 | 0.692 | 0.618 | 0.618 |
| WF | 0.890 | 0.870 | 0.477 | 0.477 |

**Table 3. HTMT Ratio.**

| Construct Relationships | HTMT Value |
|---|---|
| CAP<->AC | 0.207 |
| CEP<->AC | 0.261 |
| CEP<->CAP | 0.574 |
| CC<->AC | 0.445 |
| CC<->CAP | 0.194 |
| CC<->CEP | 0.114 |
| FO<->AC | 0.462 |
| FO<->CAP | 0.351 |
| FO<->CEP | 0.425 |
| FO<->CC | 0.223 |
| FP<->AC | 0.500 |
| FP<->CAP | 0.137 |
| FP<->CEP | 0.187 |
| FP<->CC | 0.297 |
| FP<->FO | 0.656 |
| NC<->AC | 0.577 |
| NC<->CAP | 0.121 |
| NC<->CEP | 0.070 |
| NC<->CC | 0.558 |
| NC<->FO | 0.240 |
| NC<->FP | 0.354 |
| OC<->AC | 0.901 |
| OC<->CAP | 0.214 |
| OC<->CEP | 0.170 |
| OC<->CC | 0.973 |
| OC<->FO | 0.374 |
| OC<->FP | 0.442 |
| OC<->NC | 0.960 |
| PSCP<->AC | 0.286 |
| PSCP<->CAP | 0.819 |
| PSCP<->CEP | 1.028 |
| PSCP<->CC | 0.157 |
| PSCP<->FO | 0.491 |
| PSCP<->FP | 0.214 |
| PSCP<->NC | 0.107 |
| PSCP<->OC | 0.214 |
| WLAP<->AC | 0.320 |
| WLAP<->CAP | 0.776 |
| WLAP<->CEP | 0.841 |
| WLAP<->CC | 0.181 |
| WLAP<->FO | 0.634 |
| WLAP<->FP | 0.290 |
| WLAP<->NC | 0.163 |
| WLAP<->OC | 0.254 |
| WLAP<->PSCP | 1.029 |
| WF<->AC | 0.537 |

*(Continued)*

**Table 3.** (Continued)

| Construct Relationships | HTMT Value |
|---|---|
| WF <-> CAP | 0.251 |
| WF <-> CEP | 0.317 |
| WF <-> CC | 0.296 |
| WF <-> FO | 1.037 |
| WF <-> FP | 1.057 |
| WF <-> NC | 0.341 |
| WF <-> OC | 0.459 |
| WF <-> PSCP | 0.365 |
| WF <-> WLAP | 0.480 |

As can be seen in Table 4 above, the validation results for the higher-order constructs reveal that some lower-order dimensions significantly contribute to their respective higher-order constructs, while others do not. Specifically, Affirmative Commitment (β = 0.907, t = 9.491, p < 0.001), Friendship Opportunity (β = 0.821, t = 8.995, p < 0.001), Friendship Prevalence (β = 0.295, t = 2.398, p = 0.008), and Work-life Adjustment Preoccupation (β = 0.812, t = 4.663, p < 0.001) show strong, statistically significant contributions to their higher-order constructs, indicating these dimensions reliably define the constructs. In contrast, Career Adaptation Preoccupations (β = −0.020, t = 0.123, p = 0.451), Career Establishment Preoccupations (β = 0.269, t = 1.459, p = 0.072), Continuance Commitment (β = 0.189, t = 1.216, p = 0.112), and Normative Commitment (β = 0.027, t = 0.165, p = 0.434) do not significantly contribute, suggesting a weaker or non-meaningful role in defining their respective constructs within this sample. These findings indicate that while certain dimensions are key drivers of the higher-order constructs, others may require further examination or reconsideration in the conceptual model.

Table 5 shows the Variance Inflation Factor (VIF) values for all the constructs range between 1.311 and 2.148, which are well below the recommended cutoff values of 3 or 5. This indicates that there is no serious multicollinearity among the predictor variables, meaning that the constructs do not excessively overlap or correlate with each other. As a result, each construct contributes uniquely and independently to explaining the variance in the model. The low VIF values suggest that the estimates of the regression coefficients are stable and reliable, enhancing the validity of the model's findings. Overall, these VIF values confirm that multicollinearity is not a threat to the model's robustness.

The VIF values for all path relationships are well below the threshold of 3, indicating no multicollinearity issues among the predictors. This suggests that Career Preoccupation and Workplace Friendship independently and reliably contribute to explaining Organisational Commitment without redundancy and this can be envisaged in Table 6 above.

**Table 4. Validation of higher order constructs.**

| Variable Relationship | Original Sample (O) | Sample Mean (M) | Standard Deviation (STDEV) | T Statistics | P Values |
|---|---|---|---|---|---|
| Career adaptation preoccupations → Career Preoccupation | −0.020 | −0.014 | 0.165 | 0.123 | 0.451 |
| Career establishment preoccupations → Career Preoccupation | 0.269 | 0.256 | 0.184 | 1.459 | 0.072 |
| LV scores – Affirmative commitment → Organisational commitment | 0.907 | 0.889 | 0.096 | 9.491 | 0.000 |
| LV scores – Continuance Commitment → Organisational commitment | 0.189 | 0.196 | 0.155 | 1.216 | 0.112 |
| LV scores – Friendship opportunity → Work place friendship | 0.821 | 0.815 | 0.091 | 8.995 | 0.000 |
| LV scores – Friendship prevalence → Work place friendship | 0.295 | 0.295 | 0.123 | 2.398 | 0.008 |
| LV scores – Normative commitment → Organisational commitment | 0.027 | 0.020 | 0.165 | 0.165 | 0.434 |
| Work-life adjustment preoccupation → Career Preoccupation | 0.812 | 0.797 | 0.174 | 4.663 | 0.000 |

**Table 5. Outer VIF of Higher Order construct.**

| Construct | VIF |
|---|---|
| Career adaptation preoccupations | 1.583 |
| Career establishment preoccupations | 1.889 |
| LV scores – Affirmative commitment | 1.311 |
| LV scores – Continuance Commitment | 1.320 |
| LV scores – Friendship opportunity | 1.320 |
| LV scores – Friendship prevalence | 1.320 |
| LV scores – Normative commitment | 1.516 |
| Work-life adjustment preoccupation | 2.148 |

**Table 6. Inner VIF of Higher Order construct.**

| Path Relationship | VIF |
|---|---|
| Career Preoccupation → Organisational commitment | 1.259 |
| Career Preoccupation → Work place friendship | 1.000 |
| Work place friendship → Organisational commitment | 1.259 |

## Structural model

In Table 7, the outer weights results indicate the strength and significance of each indicator's contribution to its respective latent construct. All the original sample weights (O) are positive and substantial, ranging from moderate (0.500 for Continuance Commitment) to very strong (0.982 for Affirmative Commitment). The corresponding t-statistics are all well above the critical value of 1.96, and the p-values are all 0.000, demonstrating that these contributions are statistically significant at the 0.001 level. This confirms that each indicator meaningfully and reliably explains its associated construct. For example, the indicators for Career Establishment Preoccupations and Work-Life Adjustment Preoccupation strongly load onto Career Preoccupation (0.804 and 0.980, respectively), while Affirmative Commitment has a very strong loading on Organisational Commitment (0.982). Overall, these results suggest that the measurement model exhibits good indicator reliability and that the constructs are well represented by their indicators.

## Test of hypotheses

The results in Table 8, show that Career Preoccupation has an insignificant and very weak effect on Organisational Commitment ($\beta = 0.045$, $f^2 = 0.002$, $t = 0.676$, $p = 0.250$), indicating no meaningful predictive impact. In contrast, Career Preoccupation significantly and moderately predicts Workplace Friendship ($\beta = 0.454$, $f^2 = 0.259$, $t = 7.402$, $p < 0.001$), while Workplace Friendship, in turn, significantly and moderately influences Organisational Commitment ($\beta = 0.442$, $f^2 = 0.198$, $t = 6.493$, $p < 0.001$). These findings suggest that Career Preoccupation indirectly contributes to Organizational Commitment through its positive effect on Workplace Friendship.

## Mediation results

The mediation analysis indicates that Workplace Friendship significantly mediates the relationship between Career Preoccupation and Organisational Commitment, with a positive indirect effect ($\beta = 0.201$, $f^2 = 0.002$, $t = 4.884$, $p < 0.001$). This suggests that Career Preoccupation enhances Organisational Commitment primarily through its influence on Workplace Friendship and this can be seen in Table 9 above

**Table 7. Indicator Reliability of Final Model.**

| Indicator | Loading | T statistics | P values |
|---|---|---|---|
| Career adaptation preoccupations→Career Preoccupation | 0.593 | 4.817 | 0.000 |
| Career establishment preoccupations→Career Preoccupation | 0.804 | 8.704 | 0.000 |
| Work-life adjustment preoccupation→Career Preoccupation | 0.980 | 29.795 | 0.000 |
| Affirmative commitment→Organisational commitment | 0.982 | 29.478 | 0.000 |
| Continuance commitment→Organisational commitment | 0.500 | 3.847 | 0.000 |
| Normative commitment→Organisational commitment | 0.546 | 4.564 | 0.000 |
| Friendship opportunity→Workplace friendship | 0.966 | 32.563 | 0.000 |
| Friendship prevalence→Workplace friendship | 0.699 | 7.701 | 0.000 |

**Table 8. Direct path co-efficient.**

| Path | Std. Beta | f-square | T statistics | P values |
|---|---|---|---|---|
| Career Preoccupation ->Organisational commitment | 0.045 | 0.002 | 0.676 | 0.250 |
| Career Preoccupation ->Work place friendship | 0.454 | 0.259 | 7.402 | 0.000 |
| Work place friendship ->Organisational commitment | 0.442 | 0.198 | 6.493 | 0.000 |

**Table 9. Specific indirect effect.**

| Path | Std. Beta | f-square | T statistics | P values |
|---|---|---|---|---|
| Career Preoccupation ->Work place friendship ->Organizational commitment | 0.201 | 0.002 | 4.884 | 0.000 |

Table 10 shows that the predictors in the model explain a moderate proportion of variance in Organisational Commitment ($R^2 = 0.216$, adjusted $R^2 = 0.210$) and Workplace Friendship ($R^2 = 0.206$, adjusted $R^2 = 0.203$) among the participants. This indicates that the model accounts for around 21% of the changes in these constructs after controlling for other factors.

## Test of robustness

**Predictive Relevance and Power.** According to Table 11, the $Q^2$predict values show varying levels of predictive relevance, with Friendship Opportunity (0.199) demonstrating the highest relevance, while Normative Commitment (0.001) and Continuance Commitment (0.016) show low predictive relevance. In terms of prediction accuracy, Affirmative Commitment exhibits the lowest PLS-SEM RMSE (0.982) and MAE (0.780), indicating better fit, while Normative Commitment has the highest RMSE (1.002) and MAE (0.805), suggesting poorer prediction. When compared to the Linear Model (LM) and Industry Average (IA), Friendship Opportunity outperforms, showing lower RMSE values, while other constructs like Normative Commitment and Friendship Prevalence show higher errors, indicating less accurate predictions.

**Table 10. Coefficient of determination.**

| Construct | R-square | R-square adjusted |
|---|---|---|
| Organisational commitment | 0.216 | 0.210 |
| Work place friendship | 0.206 | 0.203 |

**Table 11. PLSpredict.**

| Construct | Q²predict | PLS-SEM_RMSE | PLS-SEM_MAE | LM_RMSE | LM_MAE | IA_RMSE | IA_MAE |
|---|---|---|---|---|---|---|---|
| LV scores – Affirmative commitment | 0.042 | 0.982 | 0.780 | 0.987 | 0.783 | 1.004 | 0.793 |
| LV scores – Continuance Commitment | 0.016 | 0.995 | 0.790 | 0.998 | 0.797 | 1.003 | 0.794 |
| LV scores – Normative commitment | 0.001 | 1.002 | 0.805 | 1.003 | 0.811 | 1.003 | 0.800 |
| LV scores – Friendship opportunity | 0.199 | 0.898 | 0.666 | 0.897 | 0.665 | 1.003 | 0.755 |
| LV scores – Friendship prevalence | 0.026 | 0.991 | 0.781 | 0.989 | 0.778 | 1.004 | 0.786 |

The above Fig 2 and 3 shows the models establishing a relationship among the constructs in the study.

## Importance-performance map analysis

Among these constructs, workplace friendship has emerged as the foremost predictor of organisational commitment, exhibiting the highest total effect (approximately 0.43) and superior performance level (approximately 74%). This finding indicates that workplace friendship serves as a critical driver of organisational commitment, demonstrating both significant influence and relatively high implementation effectiveness within the organisational framework. The positioning of

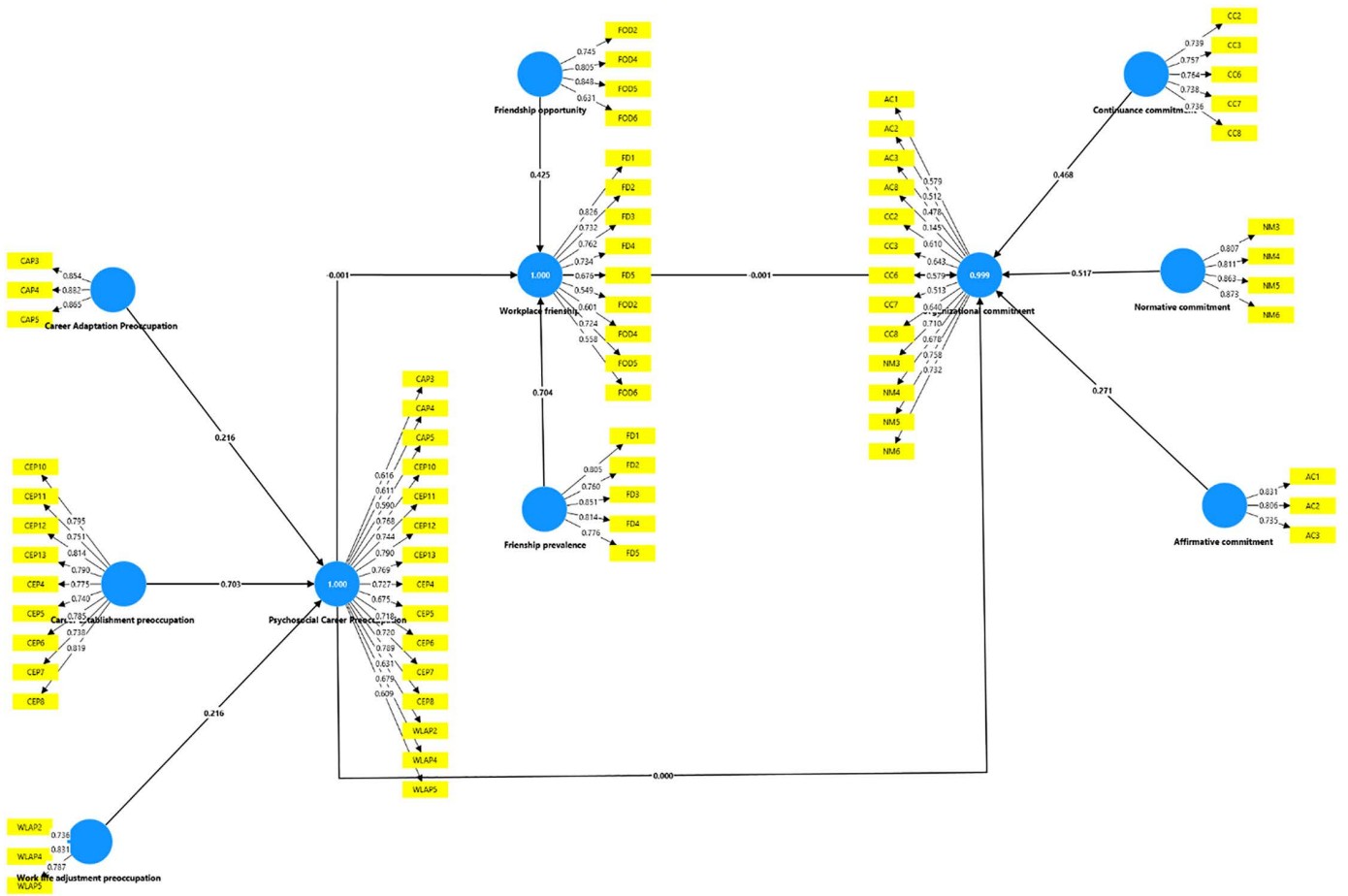

**Fig 2. Initial model.**

**Fig 3. Final model.**

workplace friendship in the high importance-high performance quadrant suggests that organisations are successfully leveraging interpersonal relationships to foster commitment, though continued investment in friendship-building initiatives could further optimize outcomes.

Career preoccupation exhibited moderate importance (approximately 0.25) but relatively lower performance (approximately 69%), establishing it as a pivotal domain requiring managerial attention. While career preoccupations demonstrate statistical significance in influencing organisational commitment, the moderate performance level indicates unexploited potential in addressing employees' career-related concerns. Strategic interventions designed to enhance career development programs, provide clearer advancement pathways, and address work-life balance issues could substantially elevate organisational commitment outcomes.

The analysis in Fig 4, reveals that workplace friendship represents the most effective lever for enhancing organisational commitment, given its dual advantage of high importance and strong performance. Conversely, career preoccupation, despite its meaningful contribution, requires targeted improvements to fully capitalize on its potential impact. Therefore, resource allocation should prioritize maintaining and strengthening workplace friendship initiatives while simultaneously implementing comprehensive career development strategies to address the performance gap in career preoccupation management.

## Discussion

This study establishes a significant empirical link between psychosocial career preoccupation and workplace friendship, which in turn substantially influences organisational commitment within Ghanaian higher education institutions. The findings reveal that career preoccupation alone has an insignificant direct effect on organisational commitment ($\beta = 0.045$, $p = 0.250$), highlighting that employees' career concerns do not translate straightforwardly into commitment without supportive relational contexts. However, career preoccupation significantly predicts workplace friendship ($\beta = 0.454$, $p < 0.001$), and workplace friendship significantly predicts organisational commitment ($\beta = 0.442$, $p < 0.001$). This indicates that psychosocial career concerns positively affect organisational commitment primarily through fostering meaningful workplace

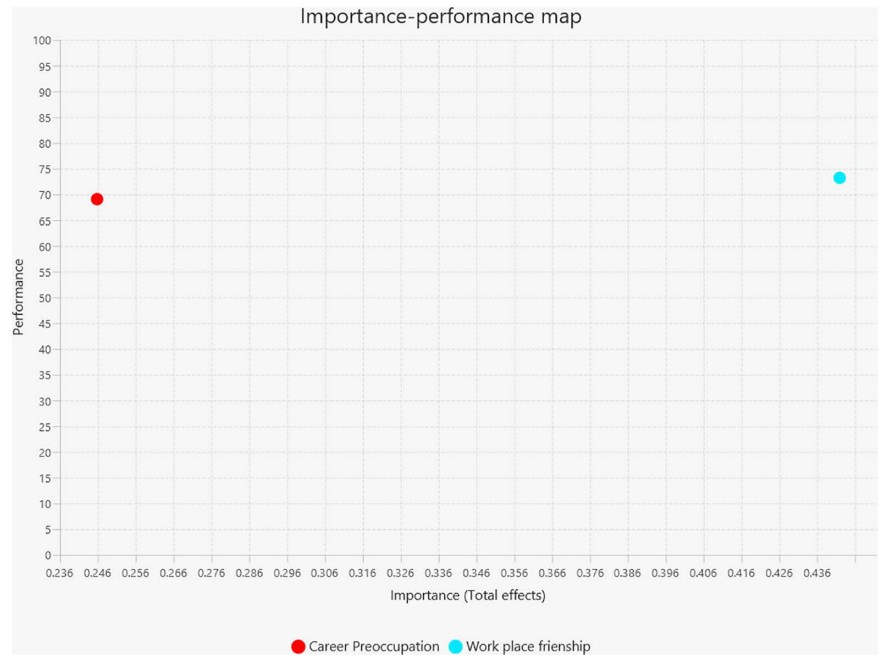

**Fig 4. IPMA-Latent.**

relationships. This finding is crucial in light of the persistent challenges faced by Ghanaian higher education institutions, such as resource constraints, staff retention difficulties, and work-life balance issues, which can strain employee attachment to their institutions [11,14].

The mediation analysis confirms that workplace friendship serves as a significant mechanism through which career preoccupation influences organisational commitment (indirect effect β = 0.201, p < 0.001). This mediating role underscores the importance of social exchange processes in organisational behavior, aligning with social exchange theory's assertion that supportive relationships fulfill employees' psychological needs and foster reciprocal commitment [26]. Furthermore, career construction theory's emphasis on adaptive behaviors in response to career concerns [25] is supported here, as employees actively use workplace friendships to navigate their career challenges, enhancing their commitment to the institution. This social and adaptive interplay is vital for sustaining faculty and staff engagement in Ghana's competitive and evolving higher education landscape.

From a practical standpoint, the findings suggest that higher education administrators should prioritize cultivating workplace friendship as a strategic resource to enhance organisational commitment. The Importance-Performance Map Analysis highlights workplace friendship as the most influential and well-implemented predictor of commitment, while career preoccupation, despite its relevance, exhibits lower performance and thus requires managerial attention to unlock its full potential. Interventions such as facilitating social interaction opportunities, encouraging collaboration, and addressing career concerns through comprehensive development programs can strengthen this relational foundation and promote greater employee attachment.

Theoretically, this study advances the understanding of organisational commitment by integrating psychosocial career preoccupation and workplace friendship within Ghanaian higher education. It extends existing empirical research, which has mostly focused on corporate settings, by providing evidence from a developing country context marked by unique structural and cultural dynamics [5,6]. The findings reconcile previous inconsistent results by revealing the indirect

pathway through workplace friendship, emphasizing the multidimensionality of career preoccupations and their complex influence on commitment. This contributes to filling a significant gap in the literature and offers a robust framework for future research and practice aimed at enhancing organisational sustainability through relational and career development mechanisms in higher education.

## Conclusions and implications

This study concludes that psychosocial career preoccupation influences organizational commitment primarily through the mediating role of workplace friendship in Ghanaian higher education institutions. While direct effects of career preoccupation on commitment are insignificant, the presence of strong, supportive workplace relationships significantly enhances employee attachment to their institutions. These findings highlight the critical importance of fostering workplace friendships as a strategic lever to translate individual career concerns into organizational loyalty.

### Managerial/practical implications

The findings underscore the importance for higher education managers and human resource practitioners to prioritize the cultivation of workplace friendships as a key strategy to enhance organisational commitment. By creating opportunities for social interaction, teamwork, and informal networking, institutions can foster an environment where employees feel supported and connected. This supportive culture not only improves job satisfaction but also acts as a conduit through which employees' career concerns are addressed more effectively, ultimately increasing their loyalty and reducing turnover intentions. Practical steps include organizing team-building activities, encouraging cross-departmental collaboration, and designing communal spaces that promote informal interactions among staff.

In addition to fostering workplace relationships, the study highlights the need for institutions to develop comprehensive career development programs that specifically respond to employees' psychosocial career preoccupations. Managers should implement clear pathways for career advancement, provide mentorship and coaching opportunities, and facilitate flexible work arrangements that support work-life balance. These initiatives can help mitigate career-related anxieties and enhance employees' sense of control over their professional growth. By addressing these career concerns proactively, institutions can boost employee engagement and motivation, which are critical for sustaining high performance and organisational effectiveness in a competitive educational landscape.

Finally, the integration of career management and workplace friendship suggests that holistic human resource strategies are essential for higher education institutions seeking long-term sustainability. Rather than treating career development and social support as separate functions, managers should adopt integrated approaches that align individual career goals with a supportive organisational culture. This could involve training supervisors to recognize and support employees' career needs within the context of fostering positive workplace relationships. Such a dual focus can create a reinforcing cycle where career satisfaction enhances social bonds and vice versa, ultimately strengthening organisational commitment and institutional resilience in the face of evolving challenges.

### Policy Implications

The study highlights the need for higher education policymakers in Ghana to develop and enforce policies that promote the integration of career development with social support systems within institutions. Policies should mandate the creation of structured programs that not only address employees' career advancement and work-life balance but also actively encourage the building of workplace friendships and peer support networks. By institutionalizing these initiatives, policymakers can ensure that higher education institutions foster environments conducive to both professional growth and interpersonal connection, which are essential for enhancing organisational commitment and retention.

Furthermore, the findings suggest that policies aimed at improving human resource management practices must consider the psychosocial dimensions of employees' career experiences. This requires the formulation of guidelines that

support flexible career pathways, mentorship programs, and employee wellness initiatives that account for employees' evolving career concerns. Policymakers should encourage institutions to adopt holistic HR frameworks that emphasize the dual importance of addressing career preoccupations and nurturing workplace relationships, ensuring that these aspects are embedded in institutional strategic plans and performance evaluation systems.

Lastly, to address systemic challenges such as bureaucratic inefficiencies and resource constraints, policy interventions should prioritize the strengthening of institutional culture that supports innovation and collaboration. This includes promoting leadership development programs that equip managers to recognize the value of workplace friendship and career support as strategic assets. Policies could also incentivize institutions to create enabling environments where employees feel psychologically safe to express their career needs and build meaningful connections. Such policy directions will be critical in enhancing the overall effectiveness and sustainability of Ghana's higher education sector in a rapidly changing global educational landscape.

## Theoretical implications

This study contributes to organisational behavior and career development theory by empirically demonstrating the mediating role of workplace friendship in the relationship between psychosocial career preoccupation and organisational commitment. This finding extends the social exchange theory (SET) by highlighting how interpersonal relationships within the workplace serve as critical channels through which individual career concerns translate into stronger organisational attachment. It reinforces the notion that reciprocal social support and trust are foundational to fostering employee commitment, especially in complex and resource-constrained environments like Ghanaian higher education institutions.

Additionally, the results support and elaborate on career construction theory (CCT) by illustrating how employees actively manage their careers through adaptive social strategies, such as cultivating workplace friendships, to navigate psychosocial career preoccupations. This dynamic interaction between individual career concerns and social context emphasizes the multidimensional and relational nature of career development, moving beyond traditional stage-based or linear models. It underscores the importance of considering social resources as integral components of career adaptability and commitment formation within organisational settings.

Moreover, this research bridges a significant gap in the literature by applying these theories within a developing country context, where organisational structures and cultural factors may differ substantially from Western-centric models. By validating SET and CCT in Ghanaian higher education, the study advances theoretical generalizability and underscores the necessity of contextualizing organisational behavior theories to reflect diverse workforce realities. This contributes to a more inclusive understanding of how psychosocial and relational factors interact to influence commitment, offering a robust framework for future research in similar emerging economies.

## Limitations and suggestions for further studies

This study acknowledges several limitations that suggest avenues for future research. First, its cross-sectional design limits the ability to infer causality between psychosocial career preoccupation, workplace friendship, and organisational commitment; longitudinal studies could better capture these dynamics over time. Second, the focus on public higher education institutions in Ghana may limit the generalizability of findings to private institutions or other sectors, so comparative studies across different organisational contexts are recommended. Again, many of this gap remain unaddressed and position their study as an initial step, with future research needed to cover the wider scope. Furthermore, the Average Variance Extracted for organisational commitment, psychosocial career preoccupation and workplace friendship were all below the threshold which shows a weak convergent validity. Additionally, reliance on self-reported data introduces potential bias, which future research could mitigate by incorporating multi-source or objective measures. Finally, while workplace friendship was identified as a key mediator, other social and organisational factors such as leadership styles or organisational justice warrant exploration to provide a more comprehensive understanding of the mechanisms influencing commitment.

## Supporting information

**S1 Data. Appendix.**

(DOCX)

## Author contributions

**Conceptualization:** Isaac Tetteh Kwao.

**Data curation:** Emmanuel Agyenim Boateng.

**Formal analysis:** Emmanuel Essandoh.

**Investigation:** Emmanuel Essandoh.

**Methodology:** Kassimu Issau, Dorothy Amfo-Antiri.

**Resources:** Esther Bema Nimo.

**Supervision:** Dorothy Amfo-Antiri.

**Visualization:** Emmanuel Agyenim Boateng.

**Writing – original draft:** Isaac Tetteh Kwao, Esther Bema Nimo.

**Writing – review & editing:** Kassimu Issau.

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
