## [Decision Letter · Decision Letter 0]

2 Sep 2025

Dear Dr. Kwao,

Thank you for submitting your manuscript to PLOS ONE. After careful consideration, we feel that it has merit but does not fully meet PLOS ONE’s publication criteria as it currently stands. Therefore, we invite you to submit a revised version of the manuscript that addresses the points raised during the review process. The comments provided by the reviewers have been appended for your perusal.

We look forward to receiving your revised manuscript.

Kind regards,

Prof. Kofi Agyekum, Ph.D.

Academic Editor

PLOS ONE

Journal Requirements:

3. Thank you for stating the following in your Competing Interests section: [We do not have competing interest].

4. In the online submission form, you indicated that [with the author and will be made available upon request.].

5. Please include captions for your Supporting Information files at the end of your manuscript, and update any in-text citations to match accordingly. Please see our Supporting Information guidelines for more information: http://journals.plos.org/plosone/s/supporting-information .

Additional Editor Comments:

Reviewer #1: See the comments below

Reviewer #2: See the comments below

Reviewers' comments:

Reviewer's Responses to Questions

**Comments to the Author**

1. Is the manuscript technically sound, and do the data support the conclusions?

Reviewer #1: Yes

Reviewer #2: Partly

2. Has the statistical analysis been performed appropriately and rigorously?

Reviewer #1: Yes

Reviewer #2: Yes

3. Have the authors made all data underlying the findings in their manuscript fully available?

Reviewer #1: Yes

Reviewer #2: Yes

4. Is the manuscript presented in an intelligible fashion and written in standard English?

Reviewer #1: Yes

Reviewer #2: Yes

Reviewer #1: This is a strong and well-structured article that makes a valuable contribution to the literature on organisational behaviour and higher education in Africa. The study is theoretically grounded in Social Exchange Theory and Career Construction Theory, and it offers empirical evidence from an underexplored context (Ghanaian higher education institutions).

The methodological rigor is commendable: the authors employed PLS-SEM with appropriate reliability and validity checks, reported detailed measurement results, and thoughtfully interpreted the findings.

The discussion and implications sections are particularly well-developed, connecting the study to Sustainable Development Goals (SDG 4, SDG 8, and SDG 10), which enhances its global relevance.

However, there are some few concerns:

1. The manuscript identifies multiple gaps in the literature, such as the multidimensionality of psychosocial career preoccupations, the varied dimensions of organisational commitment, and broader contextual challenges in Ghanaian higher education. However, the study empirically addresses only a subset of these (specifically the mediating role of workplace friendship). While this focused approach is valid, the authors should acknowledge more explicitly that many of the gaps remain unaddressed and position their study as an initial step, with future research needed to cover the wider scope.

2. Though Author(s) clearly stated that Average Variance Extracted (AVE) values for Organisational Commitment (0.363), Psychosocial Career Preoccupation (0.489), and Workplace Friendship (0.477) are below the 0.5 threshold. However, they did not state it as a limitation.

3. They discussion also, lack more empirical evidence and how it aligned with the studies' findings.

Reviewer #2: Overall, i think that the research is timely and may be very useful in addressing the issues surrounding commitment levels of staff of public institutions in Ghana. The recommendations are well on point and concise.

Below are some comments for consideration.

REVIEW COMMENTS

INTRODUCTION

1.Provide citations that support the statement made in page 10, line 8.

2.Provide citations on the research gap mentioned in page 10, line 18.

LITERATURE REVIEW AND HYPOTHESIS TESTING

1.The hypothesis stated are clear, concise and guides the study.

RESEARCH METHODOLOGY

1.The use of only one institution limits generalizability across public institutions in Ghana. Including more institution may give diverse results and better understanding on the interactions among Organizational Commitment, Psychosocial Career Preoccupation and Workplace Friendship across institutions.

2.You may want to consider control variables such as age and gender to ensure a truer relationship between the main variables of interest ie. Psychosocial career Preoccupation, Occupational Commitment and Workplace Friendship.

**Do you want your identity to be public for this peer review?** For information about this choice, including consent withdrawal, please see our Privacy Policy

Reviewer #1: No

Reviewer #2: No

---

## [Author Response · Author response to Decision Letter 1]

12 Nov 2025

Response letter

Dear editor,

S/N Reviewer(s) Responses

1 The manuscript identifies multiple gaps in the literature, such as the multidimensionality of psychosocial career preoccupations, the varied dimensions of organisational commitment, and broader contextual challenges in Ghanaian higher education. However, the study empirically addresses only a subset of these (specifically the mediating role of workplace friendship). While this focused approach is valid, the authors should acknowledge more explicitly that many of the gaps remain unaddressed and position their study as an initial step, with future research needed to cover the wider scope This has been addressed in the study. Refer to page 17 under the limitation of the study.

2 Though Author(s) clearly stated that Average Variance Extracted (AVE) values for Organisational Commitment (0.363), Psychosocial Career Preoccupation (0.489), and Workplace Friendship (0.477) are below the 0.5 threshold. However, they did not state it as a limitation.

This has been addressed in page 17 as well.

3 They discussion also, lack more empirical evidence and how it aligned with the studies' findings.

This has also been addressed.

4 The use of only one institution limits generalizability across public institutions in Ghana. Including more institution may give diverse results and better understanding on the interactions among Organizational Commitment, Psychosocial Career Preoccupation and Workplace Friendship across institutions.

This has been captured as a limitation in the study which further studies could consider.

5 You may want to consider control variables such as age and gender to ensure a truer relationship between the main variables of interest ie. Psychosocial career Preoccupation, Occupational Commitment and Workplace Friendship. This could not be addressed because the main concern of the study was the mediating role of workplace friendship. This has been well discussed in the write-up. The control variables as suggested could be considered in further studies. Again, the control variables age and gender have been extensively considered by authors; Ferreira et al. 2024.

6 Provide citations that support the statement made in page 10, line 8. This has been addressed in page 10, line 8

7 Provide citations on the research gap mentioned in page 10, line 18.

This has been addressed in page 11, line 18

---

## [Decision Letter · Decision Letter 1]

22 Dec 2025

PSYCHOSOCIAL CAREER PREOCCUPATION AND ORGANISATIONAL COMMITMENT AT HIGHER EDUCATIONAL INSTITUTIONS IN GHANA: THE ROLE OF WORKPLACE FRIENDSHIP

PONE-D-25-40156R1

Dear Dr. Kwao,

We’re pleased to inform you that your manuscript has been judged scientifically suitable for publication and will be formally accepted for publication once it meets all outstanding technical requirements.

Kind regards,

Kofi Agyekum, Ph.D.

Academic Editor

PLOS One

Additional Editor Comments (optional):

Dear Dr. Kwao,

Thank you for considering PONE for your manuscript. After the second round of reviewing, the reviewers agree that the manuscript is now ready to be considered

for publication. Congratulations and thanks once again.

Reviewers' comments:

Reviewer's Responses to Questions

**Comments to the Author**

Reviewer #1: All comments have been addressed

Reviewer #2: All comments have been addressed

2. Is the manuscript technically sound, and do the data support the conclusions?

Reviewer #1: Yes

Reviewer #2: Yes

3. Has the statistical analysis been performed appropriately and rigorously?

Reviewer #1: Yes

Reviewer #2: Yes

4. Have the authors made all data underlying the findings in their manuscript fully available?

Reviewer #1: Yes

Reviewer #2: Yes

5. Is the manuscript presented in an intelligible fashion and written in standard English?

Reviewer #1: Yes

Reviewer #2: Yes

Reviewer #1: All comments listed have been addressed in the revised manuscript, and the relevant content is clearly present. Hence the research paper can be published

Reviewer #2: Thank you for addressing all the issues raised earlier. I have checked the manuscript and it is very well improved.

**Do you want your identity to be public for this peer review?** For information about this choice, including consent withdrawal, please see our Privacy Policy

Reviewer #1: No

Reviewer #2: No

---

## [Editor Report · Acceptance letter]

PONE-D-25-40156R1

PLOS One

Dear Dr. Kwao,

I'm pleased to inform you that your manuscript has been deemed suitable for publication in PLOS One. Congratulations! Your manuscript is now being handed over to our production team.

Kind regards,

on behalf of

Prof. Kofi Agyekum

Academic Editor

PLOS One